# Identifying Individual Stressors in Geriatric Nursing Staff—A Cross-Sectional Study

**DOI:** 10.3390/ijerph16193587

**Published:** 2019-09-25

**Authors:** Bettina Wollesen, Diane Hagemann, Katharina Pabst, Ramona Schlüter, Laura L. Bischoff, Ann-Kathrin Otto, Carolin Hold, Annika Fenger

**Affiliations:** Human Movement Science, University of Hamburg, Mollerstraße 10, 20148 Hamburg, Germany; diane.hagemann@uni-hamburg.de (D.H.); katharina.pabst@studium.uni-hamburg.de (K.P.); ramona.schlueter@studium.uni-hamburg.de (R.S.); laura.bischoff@uni-hamburg.de (L.L.B.); ann-kathrin.otto@uni-hamburg.de (A.-K.O.); carolin.hold@uni-hamburg.de (C.H.); Annika.fenger@uni-hamburg.de (A.F.)

**Keywords:** stressor, stress, geriatric nursing, elderly care, housing for elderly, resilience, psychological, resilience, health risk behaviors, health behavior

## Abstract

*Background:* Nurses in geriatric care are exposed to various burdens in the workplace that result in high stress levels. The perceived stress may result in a lack of professional caring and burnout. *Objectives:* The study aim was to identify work-related and personal factors that determine stress levels to design tailored interventions. *Research design:* cross-sectional study. *Subjects:*
*N* = 195 geriatric nursing staff members. *Measures:* The survey included validated questionnaires (Trier Inventory for Chronic Stress (TICS), Health Survey Short form 12 (SF-12), Nordic Questionnaire) to identify work-related burdens, resulting stress levels and work-related behavior (AVEM). According to the limits of the Screening Subscale for chronic stress (SSCS) of the TICS, nurses were classified as stressed or non-stressed. With four-step regression analysis, main predictors for the stress level were identified. *Results:* The analysis revealed body postures, handling heavy loads, time pressure, deadlines and pressure to perform as the main burdens of the participants. Chronically stressed nurses showed different work patterns in comparison to nurses with lower stress levels. The regression analysis showed significance for the models including the work-related patterns and resilience aspects (step three (F (3.42) = 4.168; *p* = 0.010) and four (F (7.35) = 4.194; *p* = 0.002). Pattern B was a main factor for determining stress. The stress level determined the perceived burdens. *Conclusions:* Experiencing and managing stressors depends on the individual’s perception, while coping patterns—especially pattern B—can be decisive. The tailored interventions to reduce stress in geriatric nurses should focus on personal patterns.

## 1. Introduction

Occupational stress, dissatisfaction and burnout levels have increased among health personnel and may even be higher than those of other professions [1,2,3,4]. In Germany, for example, the group of health care professionals including geriatric nurses had 34.5 million days of sick leave. This is the highest rate in all industry sectors, with 8 million Euros of direct and indirect costs [5]. Internationally, a high rate of nurse absenteeism has been noted [6].

Changes in the healthcare sector associated with the increasing need of being profitable (e.g., staff-shortage) and demands related to demographic change are inducing additional burdens on the health care providing staff [7,8]. Therefore, especially geriatric nurses report increasing stress levels [9]. Stress is defined as a distress, a sense of diminished effectiveness, and is associated with reduced motivation and the development of dysfunctional attitudes and behaviors [10]. Moreover, stress results from a mismatch between intentions and the reality at the workplace [4].

The main sources of occupational stress in geriatric nurses are workload, leadership style, emotional costs of caring as well as a lack of reward, and shift-work [1]. Mental and emotional burdens are often mentioned, such as time pressure, lack of staff, dealing with aggressive residents, hierarchical structures, underpayments, and low social esteem [11,12,13,14,15]. In addition, inadequate job responsibilities [16], organizational problems, such as nursing under time pressure, are a variety of discussed stress factors (stressors) in geriatric care [17].

These stressors negatively impact the health conditions of geriatric nurses and might result in burnout, psychosomatic diseases, insomnia, substance abuse or fatigue [2,11,12,13]. Moreover, increased stress levels correlate with health-impairing behaviors and in turn the development of comorbidities [18]. This often leads to a deterioration in the quality of life, but also to a lack of service that the nurses provide [4]. Additionally, stress in health personnel is associated with patient infections [19]. Since high occupational stress levels are associated with a higher risk of chronic diseases, interventions that reduce stress are needed [20,21,22,23]. This should be regarded for geriatric nurses as well. Overall, adequate strategies to cope with high stress levels are especially important to improve health and well-being of geriatric nurses and to reduce medical errors and inadequate residential care [24].

Furthermore, it is well known that psychosocial stressors in the work environment have important links to work-related musculoskeletal disorders [25]. When encountered with different stressors and work-related psychological strains, employees may experience psychological consequences like anxiety, tension and negative emotions. The effect of strain on musculoskeletal disorders may be due to increased muscular tension, affecting hormonal, circulatory and respiratory responses. These factors contribute to a high risk for musculoskeletal disorders [26].

There is some evidence that clinical expertise, a positive attitude, work-life balance, managerial leadership, social support and self-efficacy are the main resources to prevent stress in geriatric nursing staff [27,28,29]. These factors might help to cope with requirements, to prevent the occurrence of stressors and to reduce their extent or their possible effects [30]. In addition, a few studies indicate that personality patterns determine coping strategies [31]. Personality-specific patterns of work-related behavior and experience allow the inference of both health-promoting and health-endangering stress conditions [32]. Schaarschmidt et al. identified four patterns of work-related behavior and experience (AVEM):

Pattern G: This pattern is an expression of a health promoting relationship to work. The professional ambition, the subjective importance of the work, the willingness to work and the pursuit of perfection are moderately to slightly increased.

Pattern S: This pattern is characterized by the conservational relationship to work, which can indicate dissatisfaction with the tasks and challenges at work. Often insufficient commitment to work can be observed, while there are hardly other peculiarities.

Risk pattern A: The characteristic for Pattern A is that major efforts at work are not correlated with a positive attitude towards life. Particularly noteworthy is the tremendous dedication to the job, but little resilience to stress and rather negative emotions. Health is at risk because of self-induced excessive demands.

Risk pattern B: The group of persons with the risk pattern B shows a low willingness to work. The degree of professional success, life satisfaction and social support are low. The persons do not show offensive coping strategies with the problems, but rather resignation, motivation problems as well as a decreased resistance to burdens. The described characteristics can be symptoms of a burnout syndrome [32]. For example, AVEM-risk patterns are associated with effort-reward-imbalance, chronic stress and reduced mental health of psychotherapy trainees [33].

Individual health behavior can be a positive resource to prevent stress and can therefore influence the coping processes [34]. Individual health behavior may include: the amount of sport, satisfaction with one’s own health status, the perceived importance to improve one’s own state of health, nutritional behavior, relaxing times and energy status.

So far, there is minimal research concerning stressors in the specific sector of geriatric nursing staff. Research regarding health personnel has mainly focused on cognitive-behavioral training interventions, mental and physical relaxation or organizational interventions to prevent stress at work. Still, there is a lack of specific evidence for the influence that different coping types have on the occurrence of stress in geriatric nursing staff. However, it has been evaluated as the most burdened occupational group [8]. Available data are only partially usable or transferable to our research question, because of the underlying study design, sample size or low quality of studies [12,14,35].

Consequently, the need to identify the stress levels and the high impact stressors, especially in geriatric nursing staff is urgent. Therefore, the aim of this cross-sectional study is to identify stressors that could have a negative impact on the health of geriatric nursing staff. Without identifying the main interactions of work-related stressors and the individuals’ behavior as well as their coping strategies and resources, effective interventions to prevent stress-induced health problems cannot be developed [4].

To conduct tailored interventions in the future, the main research questions of this study are: (1) Which factors (physical and mental strains and personality-specific patterns of work-related behavior) can be identified as work-related burdens in geriatric nursing staff? (2) Which factors (e.g., personality aspects or resources) determine the resulting stress levels and how?

The underlying hypothesis of the study is as follows: The working conditions and resulting physical and mental burdens cause stress among the geriatric nursing staff.

## 2. Materials and Methods

### 2.1. Study Design

This cross-sectional study analyzed data from a quantitative survey based on questionnaires from two national study projects on workplace health promotion in geriatric care from 2017/18. Therefore, this study used the method of secondary data analysis. This procedure of the study design study was approved by the local ethics committee (file reference: AZ2018_168).

### 2.2. Sample and Setting

The present analysis is based on personal data of a sample of health professionals in geriatric care facilities in Hamburg and Bremen. The data were collected in the context of two projects in nursing homes facilities at the University of Hamburg (“Netzwerk Pflege 4.0“ and “PROCARE- prevention and occupational health in long-term care”). The main purpose of the data collection for both projects was to develop health promotion interventions and reduce the sickness rate through occupational health management in the care of the elderly. Therefore, the questionnaires were used as a risk and needs assessment. Both projects were designed as special interventions to reduce health burdens. One project focused on workplace-related aspects (PROCARE) in combination with personality traits, while the other project also raised requests for educational interventions (Netzwerk Pflege 4.0). Both questionnaires therefore differed only in the second part, in the area of additional requests.

According to this procedure, the data that were analyzed in this study were collected from April 2017 to January 2018. The set of data is from a sample of health workers from 20 different nursing home facilities (in the northern part of Germany). The setting consists of facilities with and without day care and includes rural and urban businesses. Inclusion criteria were all geriatric nurses working in the examined facilities. Other employees, e.g., home management, nursing students, and kitchen staff, were excluded.

Participants were divided into two groups: stressed and non-stressed geriatric nursing staff. This assignment was operationalized by a cut-off point, which separated the stressed and non-stressed persons from each other. The cut-off point (>12 = stressed, >15 = chronically stressed) is defined by the chronic stress Screening Subscale (SSCS) of the TICS questionnaire (Trier Inventory for Chronic Stress) [36].

### 2.3. Measures

The questionnaire included six different standardized questionnaires. The questions addressed:(1)workplace exposure (Questionnaire for Subjective Assessment of Workplace Exposure- modified Slesina Questionnaire) [37];(2)musculoskeletal complaints (Questionnaire on Musculoskeletal Complaints-Nordic Questionnaire) [38];(3)physical and mental well-being (Health Survey SF-12): The SF-12 is an economic short form of the SF-36, consisting of twelve items, to measure the health-related quality of life. Its items are summarized in a sum score for mental and a sum score for physical well-being. The reliability and validity scales are considered established. Cronbachs Alpha ranged between 0.57 to 0.94 [39];(4)Trier Inventory for Chronic Stress (TICS): The twelve-item screening subscale (SSCS) of the TICS provides information about perceived stress within the last three months. Internal consistency (Cronbachs Alpha) with a range from 0.84 to 0.91 indicates good to very good reliability [36];(5)work-related behavior and experience patterns (Questionnaire on Work-Related Behavior and Experience Patterns (AVEM) [32];(6)questions regarding health-related resources due to WHO-criteria for e.g., physical activity and nutrition behavior.

In addition, personal data were collected (gender, age, occupational group). Only surveys that were completely filled out were included in the data analysis.

### 2.4. Procedures

The questionnaires were distributed among the employees in nursing homes for elderly care as a paper-pencil-version. Nursing home facilities were randomly searched and contacted by phone with the request for voluntary participation. Information about the study was given to the management of the facility or within information workshops conducted by the research team. Participants were allowed to fill in the questionnaires during working hours. Questionnaires were deposited for non-present employees of the company. The participation was voluntary. All participants were informed about the objective, the framework, the background of the data collection and the processing of the personal data by a detailed information sheet. Every participant gave his/her consent to participate by referring to a privacy policy that was attached to the consent form at the beginning of the questionnaire. The pseudonymization was done by using a coding procedure.

### 2.5. Analysis

Data evaluation was conducted in three steps.
The whole group descriptive analysis and frequencies of relevant physical and psychological work-related stressors and main musculoskeletal disorders as well as resilience factors were done with Chi^2^-tests.The resulting factors from step one were analyzed by comparing groups with chronic and non-chronic stress. In addition to the Chi^2^-tests, a one-way Analysis of Variance (one-way ANOVA) was conducted (e.g., Scores of SF-12).To further analyze the potential influential factors, a Pearson product moment correlation was computed in order to prepare a four-step hierarchical regression analysis: the regression analysis was conducted to identify which variables determine stress scores of the TICS. The potential stressors identified from step 1 were divided into the following blocks for the regression analysis:
Block I:physical strains (heavy physical tasks, awkward posture, holding heavy loads, lifting heavy loads, physical well-being and standing);Block II:psychological strains (pressure to perform, time pressure, shift-work, deadline pressure and psychological well-being);Block III:pattern (G, S, A and B);Block IV:resilience factors (satisfaction of health condition, frequency of being calm and relaxed in the past seven days, importance to improve the state of health, days of being full of energy, nutritional behavior, frequency of having breakfast and stress level).

All analyses were done with IBM SPSS Statistics for Windows, Version 25.0. (IBM Corp., “IBM SPSS”, Armonk, NY, USA). A confidence interval of 95% was used in all analysis of data.

## 3. Results

### 3.1. Participants

In total, 405 of all 860 distributed questionnaires of the health promotion projects were completed, which is 47%; 202 were excluded as they did not match the inclusion criteria. Eight could not be evaluated due to missing data (cf. Figure 1). Thus, there were 195 surveys included in the present data analysis.

### 3.2. Prevalence of Potential Stressors in Geriatric Nurses

The main work-related physical stressors were heavy physical tasks (69.9%; Chi^2^ = 46.786; *p* < 0.001; C = 0.451); awkward postures (60.5%; Chi^2^ = 94.449; *p* < 0.001; C = 0.595), standing (33.9%; Chi^2^ = 9.433; *p* = 0.001; C = 0.229), and manual load handling (holding heavy loads 51.5%; Chi^2^ = 87.019; *p* ≤ 0.001; C = 0.585; lifting heavy loads 63.2%; Chi^2^ = 82.336; *p* ≤ 0.001; C = 0.570). Moreover, psychological burdens were pressure to perform (59.3%; Chi^2^ = 71.673; *p* ≤ 0.001; C = 0.623), deadline pressure (50,0%; Chi^2^ = 77.682; *p* ≤ 0.001; C = 0.637), time pressure (72.7%; Chi^2^ = 41.766; *p* ≤ 0.001; C = 0.525) and shift work (50.9%; Chi^2^ = 15.502; *p* ≤ 0.001; C = 0.349). Frequency analysis of the data revealed the following rates of musculoskeletal disorders in the surveyed nurses: the main musculoskeletal disorders in geriatric nursing staff during the last seven days were 52.4% neck complaints (Chi^2^ = 29.917; *p* ≤ 0.001; C = 0.391), 55.1% shoulder complaints (Chi^2^ = 42.458; *p* ≤ 0.001; C = 0.450), 46.6% upper back complaints/thoracic spine (Chi^2^ = 44.934; *p* ≤ 0.001; C = 0.505), and 49.3% lower back/lumbar complaints (Chi^2^ = 35.004; *p* ≤ 0.001; C = 0.435). Moreover, relevant significant resources that differed between participants with high and low stress levels were Satisfaction with health status (Chi^2^ = 34.174; *p* ≤ 0.001; C = 0.398), importance to improve the state of health (Chi^2^ = 9.568; *p* = 0.048; C = 0.223), nutritional behavior (Chi^2^ = 9.818; *p* = 0.044; C = 0.223), frequency of having breakfast (Chi^2^ = 12.420; *p* = 0.014; C = 0.251), frequency of being calm and relaxed (Chi^2^ = 30.702; *p* < 0.001; C = 0.376), frequency of full energy (Chi^2^ = 19,059; *p* = 0.001; C = 0.306) and the self-assessed stress level (Chi^2^ = 28.431; *p* ≤ 0.001; C = 0.379; cf. Table 1).

### 3.3. Comparison of the Geriatric Nurses with Chronic and Non-Chronic Stress

The comparison of the geriatric nurses with chronic and non-chronic stress relating the resulting factors from step one is summarized in Table 1.

The results of the multi-factorial analysis of variance showed that stressed and non-stressed nurses differed in their physical as well as their psychological well-being (physical: F (1.171) = 16.550, *p* ≤ 0.001, *p*Eta^2^ = 0.088; psychological: F (1.121) = 68.838, *p* ≤ 0.001, Eta^2^ = 0.287, SSC-value F (1.194) = 318.621, *p* ≤ 0.006, *p*Eta^2^ = 0.623 and in pattern G: F (1.93) = 7.902, *p* ≤ 0.001, *p*Eta^2^ = 0.079, pattern S: F (1.93) = 2.927, *p* ≤ 0.090, *p*Eta^2^ = 0.031 and pattern B: F (1.93) = 12.964, *p* ≤ 0.001, *p*Eta^2^ = 0.124 but not in pattern A: F (1.93) = 3.685, *p* ≤ 0.058, *p*Eta^2^ = 0.039). Therefore, non-stressed geriatric nurses had higher values in physical well-being, psychological well-being, pattern G and pattern S in comparison to stressed geriatric nurses. Stressed subjects showed higher values in the SSCS-score as well as in the (risk-) pattern A and B.

### 3.4. Analysis of Correlation

According to the correlation matrix there were significant negative relations between the stress-scores and heavy physical tasks (r = −0.39), awkward postures posture (r = −0.30), holding heavy loads (r = −0.34), lifting heavy loads (r = −0.33), pressure to perform (r = −0.38), shift work (r = −0.25), and time pressure (r = −0.35) (Appendix A, Table A1). In subjects with higher stress levels these work-related burdens were higher than in subjects without chronic stress.

Furthermore, we found negative correlations between the following factors and the SSCS-values: physical well-being (r = −0.42), mental well-being (r = −0.64), pattern G (r = 0.23), pattern S (r = −0.39), pattern A (r = 0.41) and pattern B (r = 0.55) (Appendix B, Table A2). Hence, participants with high stress values had lower physical and mental well-being and more often showed pattern G, A and B.

The correlation matrix of the resilience-factors (resources) revealed that the SSCS-values correlated significantly with the satisfaction of health status (r = 0.487) and the frequency of being calm and relaxed (r = 0.49) (Appendix C, Table A3). For these aspects it was considered that the more unfavorable the behavior of the subjects, the higher the stress values.

There were no significant correlations between the SSCS-values and the physical burdens (Appendix D, Table A4).

### 3.5. Results of the Regression Model

All steps of the regression analysis showed significance for the overall model (cf. Table 2) with the highest R^2^ for step three (F (3.42) = 4.168; *p* = 0.010) and four (F (7.35) = 4.194; *p* = 0.002; cf. Table 2).

Step one to three revealed a significant effect on the physical well-being. Participants with lower physical well-being had higher SSCS-values.

Moreover, in steps two and three, a significant effect for the mental well-being was found: the increasing SSCS-values correlate with reduced well-being.

Steps three and four showed significant effects for “standing”, “time pressure”, and “pressure to perform”. Furthermore, the work-related behavior and experience patterns—especially pattern B—were significant in both steps (*p* < 0.05). Subjects with (risk-) pattern B showed had higher SSCS-values. Pattern A was only significant in step three (*p* < 0.05).

## 4. Discussion

The main aim of this cross-sectional study was to identify the interaction of work-related burdens and stress levels of geriatric nursing staff to conduct tailored interventions. Overall, the physical and mental well-being was shown to have an impact on the perceived stress levels. Moreover, the analysis identified patterns of work-related behavior and experience as relevant aspects.

Previous study results mostly suggested individual external psychological factors as possible stressors in nurses [1,15,17]. For example, McVicar revealed shift work as a possible stressor in geriatric nursing [1]. This aspect was not supported with the data of our regression model with regard to chronic stress. Nevertheless, this aspect was reported as a burden by the participants of this study with high stress levels (cf. Table 1).

For the examined factor “time pressure”, our study confirmed previous results of Nübling et al. [17] and Cope et al [15]. Time pressure mostly results from staff shortage and organizational factors. Therefore, the results of this study support the demand of changing the staff ratios to reduce stress. [1].

In addition to psychological burdens, the results from the NEXT study identified physical burdens, e. g., carrying heavy loads as a possible work strain factor [40]. Comparing our samples of stressed vs. non-stressed nursing staff, this study showed significant differences in self-reported experience of typical work-related burdens and strains (cf. Table 1). However, these aspects failed to be significant in the regression model. Thus, these aspects might interact with the individuals’ feelings of perceived stress. Taking our study results into account, one might conclude that participants with higher stress levels rate the perceived burdens higher than participants with lower stress levels. These interaction processes, which also might be influenced by the individuals’ well-being, should be further examined in future studies and future health promotion programs.

For the prevention of stress in geriatric nursing staff, a positive attitude and work-life balance are helpful resources [27,28,29]. The individual experience of stress also depends on personal characteristics and not only on environmental factors [41]. It is well known that people react differently to the same external influences. Therefore, a certain factor, such as time pressure, can lead to negative stress in some people while motivating others.

Our study results amplify these aspects while adding the analysis of another variable: the patterns of work-related behavior. In our sample of geriatric nursing staff, “risk pattern B” was one of the most important factors of experiencing negative stress at work. According to Schaarschmidt et al. the outstanding features of people with “risk pattern B” are a high tendency to resignation, low levels in the offensive problem solving, low levels of inner peace and serenity and a lack of success in the profession [32]. People belonging to this pattern have low values in the dimensions of work engagement, in the subjective importance of work and in professional ambition. People are characterized by limited distancing ability and reduced resistance to stress. It has been suggested that pattern B plays a significant role in the development of stress and the resulting physical and mental well-being.

In summary, our study results support the assumption that personality patterns determine coping strategies as reported by Carver et al. [30]. Our regression model revealed that the patterns of work-related behavior have an impact on the stress levels and determine the weight of burdens and strains. Therefore, we have to refuse the hypothesis that “the working conditions cause stress among the geriatric nursing staff”. Regarding this aspect, future interventions should integrate and control these individual aspects. Participants with risk pattern B need additional competencies for stress reduction and coping strategies. Moreover, with regards to the results of this study, one might argue that health promotion programs as well as health assessments should integrate the identification of risk patterns. A practical implication this study offers is that it seems to be necessary to start the health promotion process for this target group with a reduction of negative risk patterns. Afterwards, other interventions like ergonomic interventions of stress management might be more successful. Moreover, this procedure could be a key for a more sustainable behavior change, especially for the people with risk pattern B who suffer from resignation as well as low levels of problem-solving strategies and inner peace.

### Strengths and Limitations

One strength of our work is the high number of participants. This is the first study examining the interaction of stressors and resources of geriatric nursing staff with a large sample size. To our knowledge, this was the first study which involved work-related behavior and experience patterns in connection with stress in geriatric nursing staff. This inclusion gives a new perspective and a basic understanding of factors that influence stress. Therefore, the results provide the baseline for demand-oriented interventions.

However, our work has also certain limitations. The data collection was carried out in the northern area of Germany. This could have resulted in a local bias. Furthermore, the participants were allowed to fill in the surveys during their working times. Possible time pressure could have influenced the quality of the answers. We did not ask and control for other sociodemographic aspects than gender, age and area of employment. Therefore, we are not able to discuss the association between the variables under different aspects than these. Nevertheless, there might be a selection bias, as only nursing staff who was motivated to fill out the questionnaires could be integrated. According to the observational character of this study, a control group was not integrated. Therefore, the results of geriatric nursing staff cannot be regarded in relation to other occupational groups. Last, the used questionnaires might not involve every potential stressor in geriatric nursing staff. A qualitative part of the questionnaire could have made it possible to identify uncommon or unknown stressors. The calculation of the sample based on the total target population was not performed because of the nature of this study as a secondary data analysis. In addition, due to the nature of the cross-sectional design, the causal interpretation of our current findings is only speculative. Future studies should adopt a randomized controlled design to better delineate the causal links between stress and possible factors of stress.

## 5. Conclusions

This study identified the relation between relevant stressors and resulting strains for geriatric nursing staff. This is relevant for the development of effective interventions to prevent stress-induced diseases [4]. So far, current interventions that aim to reduce stress in the nursing sector have responded to external and single stressor factors. Experiencing and managing stressors depend on the individual’s perception. At the same time, coping patterns—especially pattern B in elderly care—can be decisive. Our results provide the missing link to improve the conditions in elderly care. They respond to the current changes in our healthcare sector. Based on our findings it is now possible to design and conduct tailored interventions, taking different personality patterns into account.

Regarding the current shortage of geriatric nurses and the expected need of additional staff for elderly care due to the aging population, it is crucial to support the health conditions of these employees. Moreover, strategies to avoid employee turnover are necessary.

## Figures and Tables

**Figure 1 ijerph-16-03587-f001:**
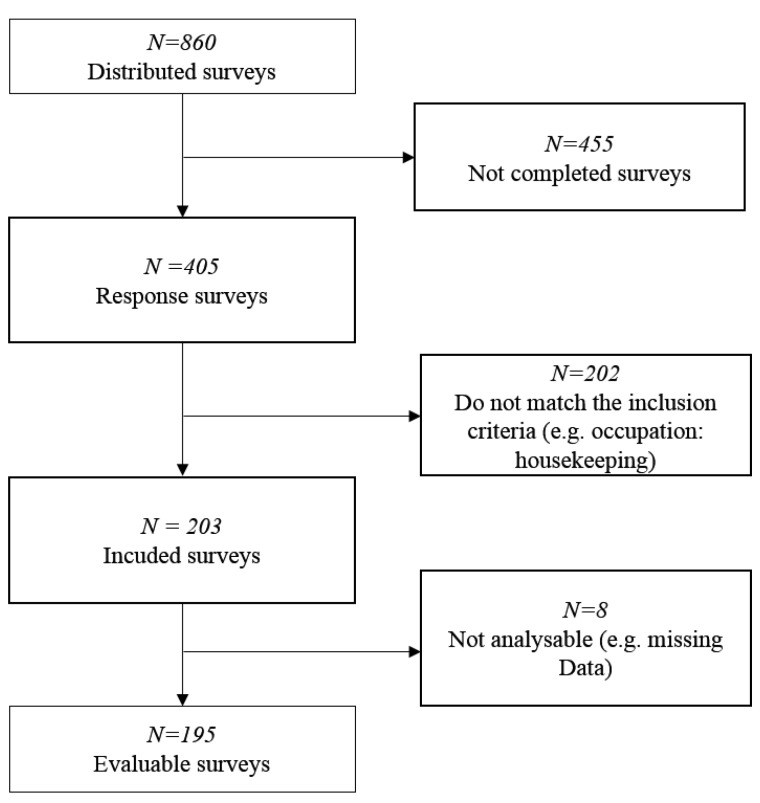
Flow diagram of survey integration process.

**Table 1 ijerph-16-03587-t001:** Characteristics of nurses with chronic stress (*n* = 124) and without chronic stress (*n* = 71).

Variables	Stressed *N* (%)	Non-stressed *N* (%)	Total *N* (%)
Gender (N) (%)	124 (64%) ^1^	71 (36%) ^2^	195 (100%)
Male Sex	14	10	24
Female Sex	107	60	167
Age	40.76 (11.7 SD)	38.97(13.1 SD)	40.1(12.2 SD)
Burden at workplace			
heavy physical tasks	100 (55.2)	32 (17.7)	132 (100)
awkward posture	80 (46.2)	32 (18.5)	112 (64.7)
standing	41 (24.0)	20 (11.7)	61 (35.7)
holding heavy loads	71 (41.3)	24 (14.0)	95 (55.2)
lifting heavy loads	82 (47.7)	32 (18.6)	114 (66.3)
pressure to perform	50 (43.9)	16 (14.0)	66 (57.9)
deadline pressure	25 (37.8)	15 (12.6)	40 (50.4)
time pressure	57 (49.6)	25 (21.7)	82 (71.3)
Shift work	42 (36.2)	18 (15.5)	60 (51.7)
Burden at the locomotor system during the last 7 days			
neck complaints	67 (41.9)	18 (11.3)	85 (53.1)
thoracic spine	47 (54.0)	17 (38.6)	64 (48.9)
lumbar spine	59 (39.3)	17 (11.3)	76 (50.7)
shoulder complaints	67 (41.6)	20 (12.4)	87 (54.0)
Resilience-Factors (resources)	Stressed mean (SD)	Non-stressed mean (SD)	Total mean (SD)
satisfaction of health status	3.31 (0.98)	2.45 (1.02)	3.0 (1.08)
Importance to improve status of health	1.85 (1.05)	2.27 (1.32)	2.0 (1.17)
nutritional behavior	2.90 (0.90)	2.75 (1.10)	2.84 (1.0)
frequency of breakfast	2.40 (1.38)	2.39 (1.54)	2.39 (1.42)
frequency of being calm and relaxed	3.20 (1.12)	2.33 (1.04)	2.89 (1.15)
frequency of being full of energy	3.33 (1.01)	2.70 (0.93)	3.10 (1.02)
stress level	3.72(1.10)	2.83 (1.16)	3.41 (1.20)
Psychological factors			
physical well-being	46.41 (9.36)	51.75 (6.14)	48.38 (8.68)
psychological well-being	41.49 (10,72)	53.47 (5,53)	45.92 (10.81)
SSCS-value	24.93 (7.37)	7.99 (4.06)	18.76 (10.36)
pattern G	19.67 (29.63)	40.08(40.59)	27.49 (35.47)
pattern S	29.36 (36.17)	43.33 (41.95)	34.71 (28.87)
pattern A	23.67 (32.29)	11.54 (25.20)	19.03 (30.22)
pattern B	27.26 (35.18)	5.03 (14.48)	18.74 (30.90)

*N* = participants. ^1^ 3 of 124 participants of the stress group did not declare their gender. ^2^ 1 of 71 participants of the not stressed group did not declare its gender.

**Table 2 ijerph-16-03587-t002:** Summary of the four step hierarchical regression analysis.

Variables	Step 1		Step 2		Step 3		Step 4	
	B	ß	B	ß	B	ß	B	ß
1. Block I: Physical burdens								
1.1 heavy physical tasks	−4.629	−0.204	0.977	0.043	2.528	0.112	4.925	0.217
1.2 awkward posture	−1.036	−0.048	−1.647	−0.076	−2.050	−0.094	−0.887	−0.041
1.3 lifting heavy loads	−4.028	−0.185	−5.404	−0.249	−2.762	−0.127	−3.582	−0.15
1.4 holding heavy loads	−0.627	−0.030	0.678	0.033	−1.294	−0.062	−3.928	−0.189
1.5 standing	0.916.	0.043	3.212	0.150	3.148	0.147	6.381 *	0.297
1.6 physical well-being	−0.461 **	−0.352	−0.372 **	−0.285	−0.354 **	−0.271	0.128	0.098
2. Block II: Psychological burdens								
2.1 mental well-being			−0.437 ***	−0.285	−0.341 ***	−0.360	0.117	0.123
2.2 pressure of time			−3.690	−0.153	−2.464	−0.10	−7.619 **	−0.316
2.3 pressure to perform			0.937	0.044	1.095	−0.05	5.279 *	0.246
2.4 deadline pressure			−1.857	−0.088	−2.679	−0.128	−2.770	−0.132
2.5 shift work			−2.908	−0.139	−1.963	−0.094	−3.728	−0.178
3. Block III: Patterns of work-related behavior and experience								
3.1 pattern A					0.080 *	0.220	0.062	0.170
3.2 pattern B					0.121 **	0.361	0.122 *	0.3363
3.3 pattern S					excluded		excluded	
3.4 pattern G					0.051	0.168	0.039	0.127
4. Block IV: Resilience factors								
4.1 satisfaction of health condition							3.463	0.37
4.2 frequency of being calm and relaxed							2.162	0.250
4.3 stress-level							0.859	0.01
4.4 frequency of having breakfast							2.32	0.202
4.5 nutritional behavior							−0.94	−0.029
4.6 frequency of being full of energy							0.295	0030
R^2^	0.376 **		0.553 ***		0.631 *		0.759 **	

Legend: B: Beta (non-standardized coefficient); ß: Beta (standardized coefficient); R^2^: R-squared coefficient of determination; *: *p* = 0.001, **: *p* = 0.01; ***: *p* = 0.0.

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
