# Peer review of "Identifying Individual Stressors in Geriatric Nursing Staff—A Cross-Sectional Study"

_ijerph, 2019, doi:10.3390/ijerph16193587_

Round 1

Reviewer 1 Report

Here are some comments that may improve some aspects:

The use of MeSH descriptors is recommended as keywords. The introduction adequately and clearly describes the phenomenon of which situations generate greater stress in geriatric nurses. It would only be interesting to highlight some similar studies that proposed and the results obtained, in addition to whether these relate to the different types of patterns. In Sample and Setiing it is necessary to explain the calculation of the sample based on the total target population indicating the precision and proportion parameters used, in addition to indicating the type of sampling In Measures it is of great interest that they provide the validation data of the scales used in the population included as the Cronbach Alpha. It is indicated that 860 questionnaires were distributed without taking into account the exclusion criteria and that is why they had to be rejected 202. It should not have been distributed to all without knowing if they met inclusion criteria as it represents a saturation situation among the participating professionals. In the results it would be interesting, after explaining the participants, to include a sociodemographic description (mean, standard deviation, frequencies, etc), before indicating the association between variables.

Author Response

We thank the reviewer for all the comments and recommendations that helped to improve the quality of the manuscipt.

Reviewer 1

The use of MeSH descriptors is recommended as keywords.

Answer: We thank the reviewer for this comment. We adjusted the list of keywords and added MeSH descriptors and also deleted those who are not useful as MeSH-Term.

Row 36-38: Keywords: stressor; stress; geriatric nursing; elderly care; housing for elderly; resilience; psychological; health risk behaviors; health behavior.

The introduction adequately and clearly describes the phenomenon of which situations generate greater stress in geriatric nurses. It would only be interesting to highlight some similar studies that proposed and the results obtained, in addition to whether these relate to the different types of patterns.

Answer: We thank the reviewer for this comment. We identified a relevant study and integrated the references into the introduction.

Row 96-98: For example, AVEM-risk patterns are associated with effort-reward-imbalance, chronic stress and reduced mental health of psychotherapy trainees’ in Germany. [34]

Grundmann J, Sude K, Löwe B, et al. Arbeitsbezogene Stressbelastung und psychische Gesundheit: Eine Befragung von Psychotherapeutinnen und therapeuten in Ausbildung. Psychotherapie, Psychosomatik, medizinische Psychologie. 2013; 63 (3-4): 145-149.

In Sample and Setting it is necessary to explain the calculation of the sample based on the total target population indicating the precision and proportion parameters used, in addition to indicating the type of sampling

Answer: We agree with the reviewer’s opinion. However, this study was a result of a secondary analysis of data. Therefore, we added this aspect into the limitation section.

Row 349- 351: The calculation of the sample based on the total target population was not performed because of the nature of this study as a secondary dada analysis.

In Measures it is of great interest that they provide the validation data of the scales used in the population included as the Cronbach Alpha.

Answer: We thank you for this comment. We have been able to add appropriate information to the questionnaires that have been validated.

Row 156-173: The questionnaire included six different standardized questionnaires. The questions addressed:

(1) workplace exposure (Questionnaire for Subjective Assessment of Workplace Exposure- modified Slesina Questionnaire) [38],

(2) musculoskeletal complaints (Questionnaire on Musculoskeletal Complaints- Nordic Questionnaire) [39],

(3) physical and mental well-being (Health Survey SF-12): The items from the SF-12 are summarized in a sum score for mental and a sum score for physical well-being. The Health Survey (SF12) is an economic short form of the SF-36, consisting of twelve items, to measure the health-related quality of life. The Scales reliability and validity are considered established. Cronbachs Alpha ranged between .57 to .94. [40]

(4) Trier Inventory for Chronic Stress (TICS): The twelve-item screening subscale (SSCS) of the TICS provides information about perceived stress within the last three months. Internal consistency (Cronbachs Alpha) with a range from .84 to .91 indicates good to very good reliability. [37],

(5) work-related behavior and experience patterns (Questionnaire on Work-Related Behavior and Experience Patterns (AVEM) [33] and

(6) questions regarding health-related resources due to WHO-criteria for e.g., physical activity and nutrition behavior.

It is indicated that 860 questionnaires were distributed without taking into account the exclusion criteria and that is why they had to be rejected 202. It should not have been distributed to all without knowing if they met inclusion criteria as it represents a saturation situation among the participating professionals.

Answer: Thanks for this advice. We made it more clearly, that our study used data which was collected by a previous study. Therefore, the exclusion criteria of our study were applied to the 860 questionnaires of the primary studies.

Row 130: Therefore, this study used the method of secondary data analysis.

Row 215:

In the results it would be interesting, after explaining the participants, to include a sociodemographic description (mean, standard deviation, frequencies, etc), before indicating the association between variables.

Answer: We thank the reviewer for this recommendation. Unfortunately, we did not ask and control for other sociodemographic aspects than gender age and area of ​​employment. We addressed this aspect in the limitation section.

Row 341-342: We did not ask and control for other sociodemographic aspects than gender and age and area of employment. Therefore, we are not able to discuss the association between the variables under different aspects then these.

Reviewer 2 Report

The paper presents the interesting topic and research to the reader.  The paper is fair to publish. 

However, it is suggested the authors to review again for the English proofreading before submitting to publication.

Author Response

Answer: We thank the reviewer for this recommendation and double-checked the language editing again.

Reviewer 3 Report

Thank you for inviting me to review this manuscript, which reports on quantitative analyses examining the impact of individual stressors in geriatric nurses, using cross-sectional data from two German national surveys on workplace health promotion in geriatric care. The paper is well written and easy to follow.

Introduction - The background is clear and well presented, as is the case for the relevance of the study reported in this paper. My only suggestion here would be to try and frame it a bit more clearly for an international readership.

Methods - The methods are generally well documented, I would only make the following three suggestions:

(1) providing a bit more detail on the parent study (i.e the two national surveys);

(2) whilst the measures are well described, prior to that, the paper should appropriately define all the variables of interest (including predictors, potential confounders and effect modifiers) which should then be related to the description of the measures (i.e. relating the description to each variable of interest);

(3) efforts to address potential sources of bias should be more clearly identified/described in the methods section.

Discussion – whilst the authors offer an appropriately balanced overall interpretation of the results, I think the discussion is still too broad and should offer a more clear direction in terms of the implications and/or meaningfulness of the study findings other than refuting the original hypothesis (i.e. what does it mean and/or what are the implications of refuting this hypothesis both in terms of the previous research and existing literature as well as in practice for geriatric nursing and/or for health organisations?).

Author Response

We thank the reviewer for all the comments and recommendations. They helped to improve the quality of the manuscript.

Answer: We thank the reviewer for this comment. We opened the focus of this study for a more international readership through language editing and added international information.

Row 45: Internationally, a high rate of nurse absenteeism has been noted. [6]

Burmeister EA, Kalisch BJ, Xie B, et al. Determinants of nurse absenteeism and intent to leave: An international study. Journal of nursing management. 2019;27(1):143-153.

Row 105: adjustments

Row 292: adjustments

Methods - The methods are generally well documented, I would only make the following three suggestions:

providing a bit more detail on the parent study (i.e the two national surveys);

Answer: We added information about the two data source studies. In reporting this section, we have also geared ourselves to STROSA (Standardized Reporting Of Secondary data Analyses).

Row 133-149: The present analysis is based on personal data of a sample of health professionals in geriatric care facilities in Hamburg and Bremen. The data was collected in the context of two projects in nursing homes facilities at the University of Hamburg ("Netzwerk Pflege 4.0 " and "PROCARE- prevention and occupational health in long-term care"). The main purpose of the data collection for both projects was to develop health promotion interventions and reduce the sickness rate through occupational health management in the care of the elderly. Therefore, the questionnaires were used as a risk and needs assessment. Both projects were designed for special interventions to reduce health burdens. One project addressed more workplace-related aspects (PROCARE) in combination with personality traits, and the other project also raised requests for educational interventions (Netzwerk Pflege 4.0). Both questionnaires therefore differed only in the second part, in the area of additional requests.

According to this procedure, the data that was analyzed in this study was collected from April 2017 to January. It contains a set of data from a sample of health workers from 20 different nursing home facilities (in the northern part of Germany). The setting consists of facilities with and without day care and includes rural and urban businesses. Inclusion criteria were all geriatric nurses working in the examined facilities. Other employees, e. g., home management, nursing students, kitchen staff were excluded.

whilst the measures are well described, prior to that, the paper should appropriately define all the variables of interest (including predictors, potential confounders and effect modifiers) which should then be related to the description of the measures (i.e. relating the description to each variable of interest)

Answer: We do not know if we understood this recommendation in the right way. The variables of interest are a result out of the whole data analyzing and reducing process during the main steps of the statistical methods until the regression analysis. The relevant correlations can be found in the Annex. We did not calculate a statistical model that allows the identification of confounders.

efforts to address potential sources of bias should be more clearly identified/described in the methods section.

Answer: We controlled the data for age and gender. Nevertheless, there might be a selection bias, as only nursing staff who was motivated to fill out the questionnaires could be integrated. Therefore, we addressed this aspect in the limitation section.

Row 343-344: We controlled the data for age and gender. Nevertheless, there might be a selection bias, as only nursing staff who was motivated to fill out the questionnaires could be integrated.

Discussion – whilst the authors offer an appropriately balanced overall interpretation of the results, I think the discussion is still too broad and should offer a more clear direction in terms of the implications and/or meaningfulness of the study findings other than refuting the original hypothesis (i.e. what does it mean and/or what are the implications of refuting this hypothesis both in terms of the previous research and existing literature as well as in practice for geriatric nursing and/or for health organisations?).

Answer: We thank the reviewer for this recommendation and added implications for practice.

Row: 321-330: Regarding this aspect, future interventions should integrate and control these individual aspects. Participants with risk pattern B need additional competencies for stress reduction and coping strategies. Moreover, with regards to the results of this study one might argue, that health promotion programs as well as health assessments should integrate the identification of risk patterns. A practical implication this study offers, is that it seems to be necessary to start the health promotion process for this target group with a reduction of negative risk patterns. Afterwards other interventions like ergonomic interventions of stress management might be more successful. Moreover, this procedure could be a key for a more sustainable behavior change, especially for the people with risk pattern B who suffer from resignation as well as low levels of problem-solving strategies and inner peace.

Round 2

Reviewer 3 Report

The authors have successfully addressed my comments. My only suggestion would be a thorough revision of English grammar and style throughout the manuscript. 

Author Response

We thank the reviewer for his recommendations. We did a third language and grammar editing with the help of a native speaker.